# The *LEPR* Gene Is Associated with Reproductive Seasonality Traits in Rasa Aragonesa Sheep

**DOI:** 10.3390/ani10122448

**Published:** 2020-12-21

**Authors:** Kenza Lakhssassi, Malena Serrano, Belén Lahoz, María Pilar Sarto, Laura Pilar Iguácel, José Folch, José Luis Alabart, Jorge Hugo Calvo

**Affiliations:** 1Centro de Investigación y Tecnología Agroalimentaria de Aragón, Instituto Agroalimentario de Aragón (IA2), CITA–Zaragoza University, 50059 Zaragoza, Spain; klakhssassi@cita-aragon.es (K.L.); blahozc@aragon.es (B.L.); mpsarto@aragon.es (M.P.S.); lpiguacel@cita-aragon.es (L.P.I.); jfolch@cita-aragon.es (J.F.); jlalabart@aragon.es (J.L.A.); 2Departamento de Mejora Genética Animal INIA, 28040 Madrid, Spain; malena@inia.es; 3ARAID, 50018 Zaragoza, Spain

**Keywords:** leptin receptor, reproductive seasonality, Rasa Aragonesa, SNP, haplotype

## Abstract

**Simple Summary:**

Seasonality of reproduction is one of the limiting factor of sheep production, with the *leptin receptor* (*LEPR*) gene associated with some reproductive traits in different species. Thereby, we searched for polymorphisms in the ovine *LEPR* gene and associated them with three reproductive seasonality traits: the total days of anoestrous (TDA) and the progesterone cycling months (P4CM), both based on blood progesterone level and related to seasonal ovarian function; and the oestrous cycling months (OCM) as an indicator of oestrous behaviour. Two non-synonymous and non-linked single nucleotide polymorphisms (SNPs) in the *LEPR* gene were involved in the OCM trait (rs403578195 and rs405459906). These findings show for the first time the involvement of *LEPR* gene in seasonality reproduction in sheep and will help to improve genetic selection programs by implementing the genotyping of reproducers, which might increase the productivity of meat sheep.

**Abstract:**

The aim of this study was to characterize and identify causative polymorphisms in the *leptin receptor* (*LEPR*) gene responsible for the seasonal variation of reproductive traits in sheep. Three reproductive seasonality traits were studied: the total days of anoestrous (TDA), the progesterone cycling months (P4CM) and the oestrous cycling months (OCM). In total, 18 SNPs were detected in 33 ewes with extreme values for TDA and OCM. Six SNPs were non-synonymous substitutions and two of them were predicted in silico as deleterious: rs596133197 and rs403578195. These polymorphisms were then validated in 239 ewes. The SNP rs403578195, located in exon 8 and leading to a change of alanine to glycine (Ala284Gly) in the extracellular domain of the protein, was associated with the OCM trait, being the G allele associated with a decrease of 12 percent of the OCM trait. Haplotype analyses also suggested the involvement of other non-synonymous SNP located in exon 20 (rs405459906). This SNP also produces an amino acid change (Lys1069Glu) in the intracellular domain of the protein and segregates independently of rs403578195. These results confirm for the first time the role of the *LEPR* gene in sheep reproductive seasonality.

## 1. Introduction

The seasonality of reproduction in sheep is a general phenomenon in breeds originating from temperate climates, such as those raised in the Mediterranean basin. Changes in the photoperiod at temperate latitudes provide the main external cue that controls the timing of out-of-season fertility [1]. In Rasa Aragonesa, an autochthonous Mediterranean sheep breed from northeastern Spain reared for meat purposes, the maximal reproductive activity is associated with short days, with the highest percentage of ewes exhibiting ovulatory activities from August to March. This reproductive seasonality induces major variation in lamb production and therefore in the market price, which suffers a decline in the price of lambs from late spring to early fall when the lamb supply is the highest. To improve the reproductive efficiency of sheep, some producers use hormonal treatments and lighting manipulation, alone or in combination, to induce ewes to lamb out of season. Both treatments efficiently induce oestrous but add expenses to producers [2]. However, the increasing demand for hormone-free products has led to the search for alternative methods, such as the introduction of rams to previously isolated anoestrous ewes to ensure the induction of ovulation and oestrous (ram effect), nutritional flushing or the use of genetic markers to select as reproducers those animals with alleles associated with an increase in out-of-season fertility. In the case of Rasa Aragonesa, Folch and Alabart [3] showed that approximately 25% of ewes have spontaneous ovulations in spring and can be naturally mated throughout the year if management conditions and nutrition are suitable. It was proven that this spring ovulatory activity is under genetic control with heritability and repeatability values of 0.20 and 0.30, respectively [4]. Seasonality is a complex trait with a strong environmental influence, expressed only in ewes, and manifested relatively late throughout life, and only in some management systems [5]. In this context, genomic approaches have been used to detect genes or genome regions influencing the ability of ewes to lamb out of season [5,6,7,8,9,10,11,12,13,14,15,16,17,18,19]. Including the selection of genotypes less sensitive to reproductive seasonality in breeding programs would be an alternative to increase the profitability and efficiency of the ovine sector.

*Leptin* gene polymorphisms have drawn much attention from animal scientists for their possible roles in economically important productive and reproductive traits [20]. In fact, leptin is primarily known for its role in the regulation of whole-body energy balance by acting on the central nervous system and influencing fat deposition in animals through the control of appetite and energy expenditure [21]. Recent experimental evidence has shown that some SNPs in the *leptin receptor* (*LEPR*) gene are associated with reproductive traits [22,23]. The physiological effects of *LEPR* on reproduction, including puberty, the oestrous cycle, pregnancy, lactation, and even the early stages of embryonic development, have been proven [24,25]. Numerous research studies have shown that leptin controls sexual maturity at the hypothalamus level [26,27,28]. Moreover, the occurrence and involvement of leptin in the hypothalamus with the release of gonadotropic hormone confirms its role in sexual maturity or reproduction [29]. Thus, leptin seems to be an important link between metabolic status and the neuroendocrine axis [30]. However, melatonin influences reproductive function via activation of receptor sites within the hypothalamic-pituitary-gonadal axis [31]. The *melatonin receptor subtype 1A* (*MTNR1A*) is considered a key gene that mediates photoperiodic reproductive seasonality in sheep [5,6,7,8,12,13,14,15,16,17,18,19]. Furthermore, expression of *LEPR* was detected in the suprachiasmatic nucleus (SCN), the mammalian “biological clock”, and the pineal gland of ruminant species [32], suggesting an interaction between photoperiod, melatonin, and leptin [1]. Although receptors for leptin and melatonin have not yet been colocalized, their presence has been demonstrated in similar hypothalamic regions in sheep [32,33,34].

*Leptin* and its receptor have been suggested as markers for enhancing productivity in livestock and are also potential candidates for marker-assisted selection [35]. In sheep, polymorphisms in *LEPR* have been associated with delayed onset of puberty and with decreased ovulation and lambing rates in prolific Davisdale sheep [36], but no studies have been performed concerning the *LEPR* gene and its involvement in seasonality reproduction in sheep. Therefore, this study aimed to identify polymorphisms in several regions of the *LEPR* gene in Rasa Aragonesa sheep and to test their association with reproductive seasonality traits.

## 2. Materials and Methods 

### 2.1. Ethics Statement

All experimental procedures were performed in accordance with the guidelines of the European Union (2003/65/CE) and Spanish regulations (RD 1201/2005, BOE 252/34367e91) for the use and care of animals in research. No hormonal treatments were applied to the ewes during the study.

### 2.2. Animal Samples

As described by Martinez-Royo et al. [11], phenotypic seasonality data were obtained from a Rasa Aragonesa sheep flock managed in an experimental farm (“Pardina de Ayés”) owned by Oviaragón S.C.L. The experimental period lasted from January to August 2012. The experimental flock was composed of 239 ewes in two age groups: 155 mature (5.2–7.2 y, 5.5 ± 0.5; mean ± SD) and 84 young (all 1.9 y, 1.9 ± 0.0) at the beginning of the experiment. Every three weeks, individual live weight (LW) and body condition score (BCS) on a 1 to 5 scale [37] were assessed. The mean LW and BCS were similar in both age groups. The pooled overall means and standard deviations for the whole experimental period were 52.5 ± 7.7 kg and 2.9 ± 0.3 for LW and BCS, respectively. Management of the ewes was described by Martinez-Royo et al. [11]; all ewes were handled in a single lot and subjected to the same management, nutrition and environmental conditions.

### 2.3. Measurement of Reproductive Seasonality Traits

Three reproductive seasonality traits were considered and described by Martinez-Royo et al. [11]. Briefly, the first one was the total days of anoestrous (TDA), based on weekly individual plasma progesterone levels. TDA was the sum of days in anoestrous, with anoestrous defined as those periods with three or more consecutive progesterone concentrations lower than 0.5 ng/mL. Likewise, ewes were not considered for this study if they were not cycling in the preceding breeding season (based on three samples taken one week apart in October), with progesterone levels under the threshold in all samples taken in January and with more than 4 consecutive samples higher than or equal to the threshold (possible pathological ewes). The second reproductive seasonality trait was the progesterone cycling months (P4CM), defined for each ewe as the rate of cycling months based on progesterone determinations. When the progesterone level was higher than or equal to 0.5 ng/mL in at least one blood sample in that month, the ewe was considered cyclic in a particular month. Finally, the third trait considered was the oestrous cycling months (OCM), defined as the rate of months cycling based on daily oestrous records for each ewe. Eight vasectomised rams fitted with harnesses and marking crayons were mixed with the ewes, and daily oestrous detection was performed [38]. Thus, after natural mating, oestrous was recorded as a colour mark on the rump of the ewes.

### 2.4. LEPR Gene Characterization

Genomic DNA was extracted from blood samples using standard protocols. The ovine *LEPR* gene is located on the chromosome OAR1, covering approximately 99 kb with 20 exons (GenBank acc. number NC_019458). Twelve exons were chosen to characterize the *LEPR* gene. These exons were selected based on having non-synonymous polymorphisms in the Ensembl Variation database (https://www.ensembl.org/info/genome/variation/index.html) on the Oar 3.1 version of the sheep genome. Primers were designed using Primer3 software version 0.4.0 (http://bioinfo.ut.ee/primer3-0.4.0/), and were designed in the intron regions around the targeted exon. The details of the oligonucleotide sequences, the annealing temperature and expected product size are summarized in Table 1. Polymerase chain reactions (PCRs) were performed in a 25 µL reaction including 25 ng of genomic DNA, 5 pmol of each primer, 200 nM dNTPs, 2.4 mM MgCl_2_, 50 mM KCl, 10 mM Tris-HCl, 0.1% Triton X-100 and 1 U Taq polymerase (Biotools, Madrid, Spain). The cycling conditions were as follows: an initial denaturation step of 94 °C for 3 min, 35 cycles of 94 °C for 30 s, annealing temperature for 30 s, and 72 °C for 30 s except for fragments 5, 6 and 11, which were 45 s, and a final extension step of 72 °C for 5 min. Direct Sanger sequencing of the PCR products from the 12 exons of 33 ewes with extreme values for TDA (low TDA: 0 days, *n* = 15; high TDA: 149.3 ± 22.3 days, *n* = 18) and OCM (low OCM: 0.24 ± 0.12, *n* = 18; high OCM: 0.88 ± 0.09, *n* = 15) were used to search for polymorphisms in the experimental population. The PCR products from genomic DNA were purified using the FlavorPrep Gel/PCR purification mini kit (Flavorgen, Ibian, Zaragoza, Spain) according to the manufacturer’s instructions. The PCR products were sequenced in both directions by STAB Vida (Caparica, Portugal) using an ABI 3730XL sequencer (Applied Biosystems, Foster City, CA, USA). The homology searches were performed using BLAST (National Centre for Biotechnology Information: https://blast.ncbi.nlm.nih.gov/Blast.cgi). For alignment of the sequences, CLUSTAL Omega (http://www.ebi.ac.uk/Tools/msa/clustalo/) and BioEdit [39] software were used. For prediction of the possible impact of an amino acid substitution on the structure and function of a protein, Variant Effect Predictor software (VEP: http://www.ensembl.org/Ovisaries/Tools/VEP?db=core) and PolyPhen-2 (http://genetics.bwh.harvard.edu/pph2/) [40] were used. Locations of SNPs were identified based on the genome version of *Ovis aries* Oar_v3.1. The secondary structure of the protein was determined from the amino acid sequence using CFSSP software (http://www.biogem.org/tool/chou-fasman/) [41].

### 2.5. LEPR Polymorphism Genotyping

Genomic DNA was extracted from blood samples of 239 ewes from the total ewes of the flock using standard protocols. Six non-synonymous SNPs were selected for genotyping the whole population: one in exon 4 (rs411478947), exon 7 (rs596133197), and exon 8 (rs403578195) and three in exon 20 (rs412929474, rs428867159, and rs405459906) (Table 2). Only non-synonymous SNPs were selected because they produce changes in the translated amino acid residue sequence and are more likely to affect the structure and function of the encoded protein and so may influence the phenotype of interest. These SNPs were genotyped by Kompetitive Allele-Specific PCR (KASP) following the manufacturer’s instructions. Sequences flanking SNPs for the SNPs were submitted for assay design to the genotyping platform provider (LGC Genomics, Biotools, Spain). For all samples, the KASP assay was carried out in a 10 μL volume containing 20 ng of genomic DNA, 5 μL of KASP V4.0 2x Master mix standard ROX (LCG Genomics, Beverly, MA, USA) and 0.14 μL of KASP-by-Design assay mix (LGC Genomics). Reactions were carried out in a CFX96 Bio-Rad thermocycler (Bio-Rad, Madrid, Spain) under the following conditions: 15 min at 94 °C followed by 9 cycles of 94 °C for 20 s and 57 °C for 1 min (dropping −0.6 °C per cycle to achieve a 55 °C annealing temperature), followed by 25 cycles of 94 °C for 20 s and 55 °C for 1 min. Following PCR, fluorescence was detected using a single quantification cycle for 1 s after cooling at 30 °C for 2 min.

### 2.6. Statistical Analysis

#### 2.6.1. SNP Association Studies

The Hardy–Weinberg equilibrium exact test was applied and the observed and expected heterozygosities and the minor allele frequency (MAF) for each SNP were calculated using Haploview software v4.2 [42]. Statistical analyses were carried out as a regression of the phenotype values of the three reproductive seasonality traits on the SNP genotypes by fitting a linear model using the Rcmdr package of R software (http://socserv.socsci.mcmaster.ca/jfox/Misc/Rcmdr/) [43]. The model included the genotype of the SNPs (S), the age (mature and young) (A), and the interaction of age by genotype of the SNPs (A × S) as fixed effects and the live weight (LW) and body condition score (BCS) as covariates. To test differences between genotypes, we estimated the least square means (LSMs) for each pairwise comparison for the SNP and SNP x age effects. A Bonferroni correction was fitted to take into account multiple tests. All SNPs were independently analysed with the same statistical model.

#### 2.6.2. Haplotype Association Studies

Blocks of linkage disequilibrium (LD) were evaluated with Haploview software v4.2 using the 4-gamete rule [42]. D’ and r^2^ within the *LEPR* were calculated and visualized in Figure 1. SNPs were phased with PLINK1.9 [44] using the expectation–maximization (E–M) algorithm to assign individual haplotypes. We considered diplotypes with a posterior probability higher than 0.7. Associations between the haplotypes and reproductive seasonality traits were performed by fitting a linear model using the Rcmdr package of R software. The model fit was similar to that used for the SNP association studies but included the haplotype (H) effect and the interaction of age by haplotype (A × H). Haplotypes for each individual were codified as 0, 1 or 2, indicating the number of copies of each haplotype. Only haplotypes with a frequency greater than or equal to 1% were considered. To test differences between haplotypes, we estimated the LSMs for each pairwise comparison. The Bonferroni correction was applied to take into account multiple tests.

## 3. Results

### 3.1. Isolation of the Partial Ovine LEPR Gene and Polymorphism Genotyping and Linkage Disequilibrium (LD)

To identify polymorphisms in the *LEPR* gene, we sequenced twelve exons that have non-synonymous polymorphisms in the Ensembl Variation database (Table 1) (https://www.ensembl.org/info/genome/variation/index.html). These exons were located at the beginning, middle and end regions of the gene. In total, 18 SNPs were detected: 3 and 8 SNPs in exons 4 (rs411478947, rs159694506 and rs159694508) and 20 (rs412929474, rs428867159, rs405459906, rs403654953, rs426037269, rs415715948, rs414501727 and rs427778198), respectively, and 1 SNP in exons 7 (rs596133197), 8 (rs403578195), 12 (rs407234698) and 16 (rs421946862) and in introns 9 (rs416296450), 10 (rs404892216) and 16 (rs401262081) (Table 2). Six of these SNPs were non-synonymous substitutions and were genotyped in the whole population, with 2 (snp_ex7 and snp_ex8) and 3 (snp_ex4, snp_ex7 and snp_ex8) of them predicted as deleterious or possibly/probably damaging by VEP or PolyPhen-2 software, respectively (Table 2). These SNPs showed low MAFs, ranging between 0.023 (snp_ex7) and 0.065 (snp_ex4) (Appendix A).

To determine the extent of LD among these markers, we estimated the parameters D’ and r^2^ between all pairwise combinations of the six non-synonymous SNPs. The results of the LD analysis are shown in Figure 1, in which two LD blocks were predicted. Block 1 is composed of SNPs located in the extracellular domain of the protein (snp_ex4, snp_ex7 and snp_ex8), and Block 2 is composed of three missense mutations located in the cytoplasmic domain of the receptor in exon 20.

### 3.2. SNP Association Studies

For the association analyses, we used 239 ewes from which thirty-five ewes (29 adults and 6 young ewes) did not present anoestrous during the experiment based on TDA trait (TDA = 0). Similarly, seventy-seven (60 adult and 17 young) and nine (7 adult and 2 young) ewes were cycling during all the experiments based on P4CM and OCM traits, respectively. All SNPs were in Hardy–Weinberg equilibrium. Only snp_ex8 showed a significant association with OCM (CC vs. GC genotypes, *p* = 0.0027). Ewes with the CC genotype showed more oestrous records (+0.12) than heterozygous ewes (Table 3). Details of type III test and LSMs for the SNP and the SNP x A effects for each SNP are provided in Appendix A. For the interaction effect between SNP and age, only the TDA phenotype differed among genotypes (*p* < 0.05) in young ewes for snp_ex20_1 after the Bonferroni correction (see Appendix A for further details). Significant difference was found between the GG genotype and AG genotype (*p* = 0.04). Indeed, young ewes with the GG genotype had higher TDA values than heterozygotes. No significant differences were detected between the AA (*n* = 2) and AG (*n* = 14) genotypes in young ewes. It is worth noting that in our population, this SNP showed a low frequency for the A allele (0.10).

### 3.3. Haplotype Association Studies

Haplotype association studies was performed taking into account the two LD blocks predicted with Haploview (Figure 1) and a block containing all SNPs (Block 0). In total, 17, 5 and 6 haplotypes were found for blocks 0, 1 and 2, respectively (Appendix A). We only considered diplotypes with a posterior probability higher than 0.7 and haplotype frequency > 0.01. Thus, haplotype analysis was conducted considering 237/8, 237/4 and 235/4 ewes/haplotypes for blocks 0 (Appendix A), 1 (Appendix A) and 2 (Appendix A), respectively.

The significant association previously found between snp_ex20_1 by age and TDA phenotype was confirmed by haplotype association studies. In this sense, young ewes with no copies of the h1 (**A**TG) haplotype (in bold, snp_ex20_1) in block 2 had higher TDA values than those with 1 copy (Table 4). Moreover, in block 0, a significant effect was found for the h1 (GCC**A**TG) haplotype containing allele A of the snp_ex20_1, showing that young ewes with one copy of h1 had more oestrous events than those without copies (Table 4).

Haplotypes h2 and h8 of block 0 were also associated with OCM considering the whole population. Ewes with 0 copies of the h2 (GC**C**GT**G**) or h8 (GC**G**AT**G**) haplotypes (in bold, snp_ex8 and snp_ex20_3) showed more oestrous records than those with 1 copy. Notably, the h8 haplotype has the G allele (deleterious allele) of snp_ex8. Similarly, the analysis of block 1 showed that haplotype h1 (GC**G**), which also contains the G allele of snp_ex8, is associated with OCM. However, ewes with haplotype h2 for block 0 (GCCGT**G**) and block 2 (GT**G**) carry the G allele of snp_ex20_3, which was associated with less oestrous records (OCM), although this SNP did not show a significant *p*-value after Bonferroni correction in SNP association studies.

## 4. Discussion

We detected 18 SNPs in the *LEPR* gene using 33 ewes with extreme values of TDA and OCM; six were non-synonymous substitutions, that were validated in 239 ewes. SIFT values varying from 0 to 1 were predicted for these SNPs by VEP software. SIFT scores lower than 0.05 suggest a potential intolerable amino acid substitution and a potential influence on protein function. In exon 4, a non-synonymous polymorphism (snp_ex4) promoting a change of arginine to cysteine at position 62 (according to their positions in GenBank acc. number ENSOARP00000011154) was detected, with this substitution predicted as tolerated but with a low SIFT value (0.05) and possibly damaging by VEP and PolyPhen-2 software, respectively (Table 2). Furthermore, arginine is a positively charged amino acid, whereas cysteine is polar in nature. This SNP was previously described by Haldar et al. [45] in Davisdale ewes. Of particular interest were two non-synonymous SNPs located in exons 7 and 8 and predicted as deleterious (SIFT = 0) by VEP analysis. The first SNP in exon 7 (snp_ex7) produces an amino acid change from threonine (polar) to isoleucine (non-polar) at position 248, whereas the second SNP in exon 8 (snp_ex8) produces a change from alanine (non-polar) to glycine (non-polar) at position 284. These three SNPs found in exons 4, 7 and 8 are located in the extracellular domain of the protein, where different amino acid substitutions have been associated with obesity in humans [46]. The three SNPs found in exon 20 were located in the cytoplasmic domain of the receptor. Two of them (snp_ex20_2 and snp_ex20_3) were previously described by Haldar et al. [45]. None of the three non-synonymous substitutions found in exon 20 were predicted as deleterious, being considered tolerated or benign, with SIFT values ranging from 0.5 (snp_ex20_1) to 1 (snp_ex20_3).

We also studied whether these mutations alter the secondary structure of the protein. In fact, four of these mutations alter the predicted secondary structure of the mature protein. Snp_ex4 promotes a putative loss of two turns, which increases the length of the random coil structure in two amino acids. Snp_ex7 and snp_ex8 putatively change one alpha helix motif by a β-pleated sheet and a turn motif, respectively. Finally, snp_ex20_3 should promote a change of a random coil by an alpha helix motif.

SNP association analysis showed that the non-conservative SNPs found in exons 4 and 7 were not associated with reproductive seasonality traits. These SNPs were predicted as tolerated (but with a low SIFT value) and deleterious, respectively, but they showed low MAF values (0.06 and 0.02 for snp_ex4 and snp_ex7, respectively) (Appendix A). Only one homozygous and no animals were found for the predicted tolerated and deleterious alleles (T alleles for both SNPs) of snp_ex4 and snp_ex7, respectively. However, Haldar et al. [45] found a strong association between snp_ex4 and puberty phenotypes (*p* < 0.001) but found a higher frequency for the T allele (0.47). These researchers reported that ewe lambs homozygous for the T allele in the *LEPR* gene were less likely to attain puberty at 1 year of age than those that did not carry the mutation in Davisdale sheep. Therefore, statistically significant effects were found concerning OCM and the deleterious SNP mutation in exon 8, showing a low MAF (0.06) and no homozygous animals for the putative deleterious allele. Heterozygous animals for this SNP showed fewer oestrous records (OCM trait) than homozygous animals. As OCM indicates behavioural signs of oestrous, it could be inferred that natural selection against homozygous animals for the deleterious allele has led to a low frequency of this allele. Haplotype association analysis confirmed these results. In fact, ewes whose haplotype contains the G allele (deleterious mutation) for snp_ex8 showed less oestrous events.

The interaction between snp_ex20_1 and age affected TDA in young ewes. However, the opposite effect was found in adult ewes (Appendix A). This finding could indicate that this mutation is not responsible for the observed effect but could be in LD with some causative mutation. In this sense, snp_ex20_1 was in LD with snp_ex20_2 (r^2^ = 0.51) and snp_ex20_3 (r^2^ = 0.42) in the predicted haplotype block 2 (Figure 1). Moreover, ewes with haplotype h2 for block 0 (GCCGT**G**) and block 2 (GT**G**), carrying the G allele of snp_ex20_3, had significantly lower OCM values after Bonferroni correction, indicating the putative involvement of this SNP of the *LEPR* gene on the seasonal phenotypes. The SNPs at exon 20 segregate independently from those located in the extracellular domain, and different effects in two different regions of the LEPR protein were found in this study. One of them, snp_ex8, is located in the CRHI/immunoglobulin-like domain of the extracellular domain of the protein, where different amino acid substitutions have been associated with obesity and disrupted pubertal development in humans [46]. The second mutation, snp_ex20_3, was not associated with puberty traits in the work of Haldar et al. [45]. This mutation was not predicted as deleterious and was located in the cytoplasmic domain close to a conserved region (called box 3) around position 1079 in the amino acid sequence. In humans, multiple splice variants of *LEPR* mRNA have been identified encoding an identical ligand binding domain but differing in the length of the cytosolic domain [47,48]. The LEPR isoforms A, B, C, and D have the same JAK binding motif encoded by exon 17. However, only the LEPR-B isoform contains the Box 3 motif encoded by exon 20 for STAT activation [47,49,50]. This isoform is expressed ubiquitously and constitutes up to 35% of the *LEPR* transcripts in the hypothalamus [51]. Only the full-length LEPR isoform (LEPR-B isoform) is able to fully transduce an activating JAK/STAT signal into the cell. Remarkably, the intracellular domain of the B isoform contains three tyrosine residues (Y986, Y1079 and Y1141) that activate the intracellular STAT signal transduction pathway. Y1079 plays a dominant role in activating STAT5, and Y1141 activates STAT3 [46]. Then, snp_ex20_3 (Lys1069Glu) could modify the STAT5 binding motif and disrupt the JAK/STAT signalling pathway. However, in Y1138S LEPR-B mutant females in mice, this mutation induced impaired STAT3 signalling with residual STAT5 function, but it did not cause infertility [52]. In livestock species, Almeida et al. [53] investigated the SNP (T945M) polymorphism in exon 20 of the *LEPR* gene in Angus, Brangus and Charolais cattle and found no associations with reproductive characteristics. The authors reported that blood leptin levels were influenced by this *LEPR* SNP in late pregnancy but not during lactation. It is important to note that associations between a mutation and the observed phenotypes are not direct evidence that the mutations caused the observed changes in phenotype. The observed relationship could indicate that the SNP is in linkage disequilibrium with the true causative mutation [36]. In our study, we used a small sample size of ewes (*n* = 33) with extreme values for reproductive seasonality traits to look for polymorphisms that could be segregating in this population. This design could increase the power to detect polymorphic SNPs associated with the trait but minimized the probability of detecting other polymorphic SNPs. Furthermore, we did not sequence the complete coding region or regulatory regions, such as the promoter or 5′ and 3′ UTRs.

Although the relationships between individuals in the sample were unknown and then population stratification cannot be checked, the results with the SNP located in exon 8 are very consistent. This consistency is justified by the significant association with the OMC trait (in SNP and haplotype association analysis), the in silico prediction of the functional and structural consequences of this non-synonymous SNP (predicted as deleterious and affecting the secondary structure of the protein) and the location in the extracellular domain of the protein, where different amino acid substitutions have been associated with phenotype effects.

In summary, these results confirmed for the first time the involvement of the *LEPR* gene in reproductive seasonality. In this sense, several studies suggest that *LEPR* influences GnRH neuron activity and GnRH secretion by crosstalk with kisspeptin [54]. Kisspeptin cells are determinants of GnRH/LH secretion in the different seasons and are responsible for activation of reproductive function. Clarke et al. [55] reported that kisspeptin expression in the arcuate nucleus is markedly reduced during the nonbreeding period and increased in ewes exposed to a short photoperiod and in the follicular phase of the cycle in the breeding season, suggesting the involvement of kisspeptin neurons in this activation. Kisspeptin neurons regulate GnRH neurons and transmit sex-steroid feedback to the reproductive axis (the trigger of increased LH secretion and gonadal activation), whereas a negative feedback of oestrogen on GnRH secretion is characteristic of the nonbreeding season [55]. This finding is comparable to that reported by other authors [56,57] about pubertal development events. In knockout mice and individuals with impaired LEPR function, disruption of pubertal development was found, as in the case of Davisdale ewes [45].

## 5. Conclusions

In conclusion, one SNP predicted as deleterious located in the extracellular domain of the *LEPR* gene (snp_ex8) was strongly associated with the oestrous cycling months in Rasa Aragonesa sheep, confirming for the first time the role of the *LEPR* gene in reproductive seasonality in ruminants. Furthermore, another non-linked SNP in exon 20 was associated with this trait, as shown in the haplotype association analysis. This SNP could be in linkage disequilibrium with other SNPs not detected in this study. The G alleles of snp_ex8 and snp_ex20_3 are associated with higher OCM values, which indicate behavioural signs of oestrous in the Rasa Aragonesa breed. Genetic selection programs can be enhanced by implementing the genotyping of reproducers for these alleles related to reproductive seasonality, which might increase the productivity of meat sheep.

## Figures and Tables

**Figure 1 animals-10-02448-f001:**
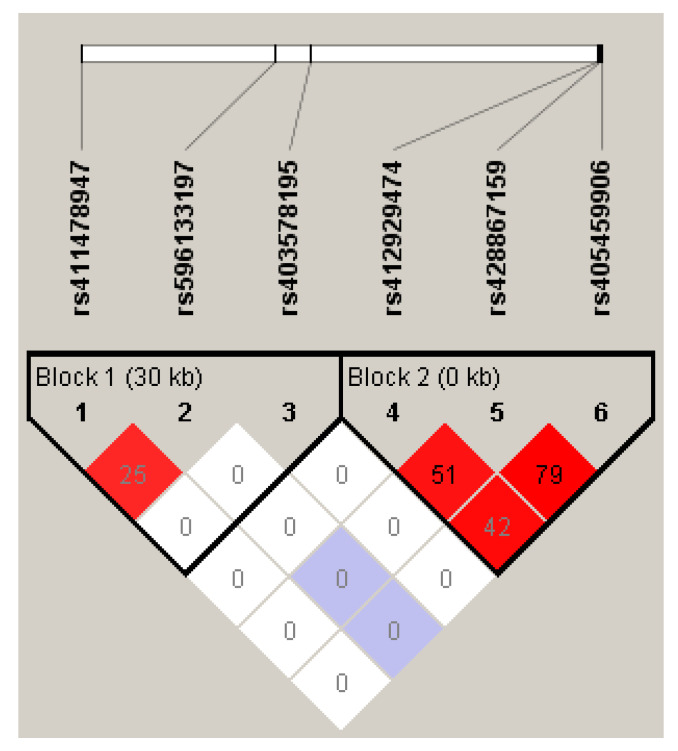
Linkage disequilibrium plot among the six non-conservative SNPs in *LEPR* using Haploview. The linkage disequilibrium colour scheme and values correspond with the D′ and r^2^ parameters, respectively. Strong LD (D′ = 1, LOD ≥ 2) is indicated in red. Red indicates varying degrees of LD with lighter shades displaying less than darker shades (D′ < 1, LOD ≥ 2), and white indicates low LD (D′ < 1, LOD < 2).

**Table 1 animals-10-02448-t001:** Primer sequences, location, annealing temperatures and amplification fragment sizes.

PCR	Primer Sequence (5′–3′) ^1^	Site	^2^ AT (°C)	Size (bp)
1	F: TTTTTCTGTGTCTTTTGAATGTCC R: AAGTAACAACTAATGCTTGGAACA	Exon 4	57	397
2	F: GCTCTTTAAGCTGGGTGTGC R: TTCAGCCTGTTTGAATGACTG	Exon 6	55	386
3	F: TGCTAAAAATTCATTTTGACTTCG R: GGAGGGCATCTCACCTTTTC	Exon 7	55	293
4	F: CTGTCGCCAGCTAACTCCTC R: CCTCCTTTTGAGTTACCACCA	Exon 8	55	378
5	F: TGCCTGGTGAATCCTTTTTA R: TCTCACCATATCCACAGAAAAAT	Exons 9–10	53	700
6	F: AGAGCTGGGAATTCAGAAATG R: TCTTTTCAATCCCACTGCAA	Exon 11	53	496
7	F: CTGCTTGGCAGGTGGATT R: CAGGAGGATGTATTTTATGCCAGT	Exon 12	55	392
8	F: TGCCTACCAATGGGAAATGT R: ATGGGAGGGGTTTGAAAGAT	Exon 15	55	383
9	F: CCTGCTTTCTCTTCCTTCTTCC R: TTTTTGAAGTTTTCATTAACTGTGTT	Exon 16	55	389
10	F: CCAGTTTCAATCCATAAATCATCA R: TGGCAGCATTGTTGCTAACT	Exon 17	55	299
11	F: TGAAGCAAAACAAAACAAAACA R: ACTCTCCTAACCAATGGTGAAA	Exon 20	52	974

^1^ F: forward; R: reverse; ^2^ AT: annealing temperature.

**Table 2 animals-10-02448-t002:** Information about the location and amino acid substitution effect of the identified SNPs according to the Variant Effect Predictor and PolyPhen-2 software in the *LEPR* gene. Scores for these programs are indicated in brackets. The SNPs are ordered according to their positions in the Oar3.1 genome version (Oar3.1: GenBank acc. number NC_019458). The amino acid positions are ordered according to their positions in GenBank acc. number ENSOARP00000011154 sequence.

SNP	Alias ^1^	Location	Position in OAR Version 3.1	Nucleotide Change	Amino Acid Change	VEP (SIFT Score)	PolyPhen-2 (Score)
rs411478947	snp_ex4	Exon 4	Oar1: g.40787726	C > T	Arg62Cys	Tolerated (0.05)	Possibly damaging (0.74)
rs159694506			Oar1: g.40787782	T > C	Asn80 = ^2^	-	-
rs159694508			Oar1: g.40787821	T > C	Ser93 =	-	-
rs596133197	snp_ex7	Exon 7	Oar1: g.40813963	C > T	Thr248Ile	Deleterious (0)	Probably damaging (0.98)
rs403578195	snp_ex8	Exon 8	Oar1: g.40818703	C > G	Ala284Gly	Deleterious (0)	Possibly damaging (0.77)
rs416296450		Intron 9	Oar1: g.40825576	G > A	-	-	-
rs404892216		Intron 10	Oar1: g.40828606	A > G	-	-	-
rs407234698		Exon 12	Oar1: g.40833201	A > G	Pro561 =	-	-
rs421946862		Exon 16	Oar1: g.40840634	C > T	Ser791 =	-	-
rs401262081		Intron 16	Oar1: g.40840703	C > T	-	-	-
rs403654953		Exon 20	Oar1: g.40857538	C > T	Gly908 =	-	-
rs412929474	snp_ex20_1		Oar1: g.40857581	G > A	Val923Ile	Tolerated (0.5)	Benign (0.04)
rs426037269			Oar1: g.40857583	C > T	Val923 =	-	-
rs415715948			Oar1: g.40857634	C > T	Ala940 =	-	-
rs428867159	snp_ex20_2		Oar1: g.40857869	C > T	Pro1019Ser	Tolerated (0.77)	Benign (0.06)
rs405459906	snp_ex20_3		Oar1: g.40858019	A > G	Lys1069Glu	Tolerated (1)	Benign (0)
rs414501727			Oar1: g.40858045	C > T	Val1077 =	-	-
rs427778198			Oar1: g.40858219	G > A	Gln1135 =	-	-

^1^ Nomenclature used for each SNP in this work. ^2^ No amino acid change.

**Table 3 animals-10-02448-t003:** Type III test for the significant SNPs and SNP by age (A) effects on the *LEPR* gene using the seasonal traits from Rasa Aragonesa ewes. The least square means (LSMs) and standard errors for the SNP and the SNP x A effects on the *LEPR* gene are also shown. Only significant SNPs after Bonferroni correction are shown. Different letters indicate significant differences: ^a,b^: *p* < 0.05.

**SNP**	**Trait**	***p* Value SNP**	**A**	**SNP LSMs**		
				**CC**	**GC**	**GG**
snp_ex8	OCM	0.003	All	0.54 ± 0.01 ^a^	0.42 ± 0.03 ^b^	-
**SNP**	**Trait**	***p* Value SNP × A**	**A**	**SNP × A LSMs**		
				**AA**	**AG**	**GG**
snp_ex20_1	TDA	0.0004	Young	80.6 ± 31.09 ^a,b^	45.6 ± 11.98 ^a^	83.3 ± 6.41 ^b^

**Table 4 animals-10-02448-t004:** Type III test for the haplotype and haplotype by age (A) effects for blocks 0, 1 and 2 on the *LEPR* gene using the seasonal phenotypic data from Rasa Aragonesa ewes. The least square means (LSMs) and standard errors for the haplotype effect on the *LEPR* gene are also shown. Only significant haplotypes after Bonferroni correction are shown. Different letters indicate significant differences: ^a,b^: *p* < 0.05.

**Haplotype Block ^1^**	**Trait**	**Haplotype**	**Frequency**	***p* Value Haplotype**	**A**	**Haplotype LSMs ^2^**		
						**0 Copies**	**1 Copy**	**2 Copies**
Block 0	OCM	h2 (GCCGTG)	0.07	0.002	All	0.53 ± 0.01 ^a^	0.44 ± 0.03 ^b^	-
	OCM	h8 (GCGATG)	0.01	0.004	All	0.52 ± 0.01 ^a^	0.26 ± 0.10 ^b^	-
Block 1	OCM	h1(GCG)	0.05	0.003	All	0.55 ± 0.01 ^a^	0.42 ± 0.03 ^b^	-
Block 2	OCM	h2 (GTG)	0.07	0.002	All	0.54 ± 0.01 ^a^	0.44 ± 0.03 ^b^	-
**Haplotype Block ^1^**	**Trait**	**Haplotype**	**Frequency**	***p* Value Haplotype × A**	**A**	**Haplotype ×** **A LSMs ^2^**		
						**0 Copies**	**1 Copy**	**2 Copies**
Block 0	OCM	h1(GCCATG)	0.09	0.004	Young	0.46 ± 0.02 ^a^	0.66 ± 0.05 ^b^	0.55 ± 0.14 ^a,b^
Block 2	TDA	h1 (ATG)	0.10	0.0003	Young	83.9 ± 6.38 ^a^	46.3 ± 11.97 ^b^	81.4 ± 31.09 ^a,b^

^1^ Block 0: snp_ex4–snp_ex7–snp_ex8–snp_ex20_1–snp_ex20_2–snp_ex20_3; Block 1: snp_ex4–snp_ex7–snp_ex8; and Block2: snp_ex20_1–snp_ex20_2–snp_ex20_3. ^2^ 0 copy: LSMs and SE for 0 copies of the haplotype; 1 copy: LSMs and SE for 1 copy of the haplotype; and 2 copies: LSMs and SE for 2 copies of the haplotype.

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
