# Peer review of "The LEPR Gene Is Associated with Reproductive Seasonality Traits in Rasa Aragonesa Sheep"

_animals, 2020, doi:10.3390/ani10122448_

Round 1

Reviewer 1 Report

The manuscript has been improved according to my suggestions. I think it can be accepted in the present form.

Author Response

We thank the referee 1 for his/her positive overall evaluation of our study.

Reviewer 2 Report

Comments on the manuscript The LEPR gene is associated with reproductive seasonality traits in Rasa Aragonesa sheep

In this reviewed version, the authors only answered partially my considerations.

First, I thank the authors for the reviewed version.

Whilst I agree with the authors that phenotyping is the most difficult part of association studies. However, it is not an excuse to do not do it. You need to have an extreme association (near to be a monogenic trait) to get results that could be interesting. I believe that this is not the case.

Second, the authors of this reviewing letter suggest that reviewers should evaluate the entire manuscript and not only the experimental design. I strongly disagree with this. Discussion is not to justify why the experimental design is weak (in fact if it is weak, you can only improve it by changing the model or the n). So, I do not see the point to evaluate the discussion of a study in which the flaws that I detect cannot be corrected by “discussing it”. The time employed by the reviewers is so important as the time used by the authors.

Finally, the authors speculate on potential mechanisms based on in silico analysis. All those analyses are made mostly in different species, and therefore, they should be taken with extreme caution. Mostly in a species in which few basic experimental designs were conducted.

Regarding this new version

The low number of individuals analyzed is the cause, from my point of view, of results such as the presented.

The authors said that they discuss it but the paraph mentioned is not a discussion of advantages or disadvantages. It's only a phrase that said that extreme values could supply a low n in terms of detection power (which is not the case in some of the results provided). This should be improved.

This is particularly clear (from my point of view) in which the same genotype showed extreme differences in phenotypes (particularly in AA and AG (to a lesser extent)), suggesting that something is missing in the analysis, but also explain why different genotypes showed the same phenotype. As I mentioned above, the fact that the authors included in the discussion several lines saying why they do not make further analysis is no excuse.

Therefore, I would consider removing (or at least reduce” the importance of TDA results and discussion on the manuscript since no additional data can be provided (Or at least I believe that since experiments were conducted 8 years ago). I suggest also to be EXTREMELY clear regarding the number of animals and the possible bias on the results in the discussion.

After this major review, the manuscript could be considered for publication.

Reviewer 3 Report

The authors have adequately addressed this reviewer's concerns and with additional edits, the manuscript has improved overall in clarity.  

One minor issue that needs to be addressed is Figure 1: The authors refer to "Pink indicates varying degree of LD....". In my version of the manuscript, Figure 1 shows red, white and light blue fields, but no pink fields. Please confirm the accuracy of colors as they show up in the manuscript.

Author Response

The authors have adequately addressed this reviewer's concerns and with additional edits, the manuscript has improved overall in clarity.  

We thank the referee 3 for his/her positive overall evaluation of our study.

One minor issue that needs to be addressed is Figure 1: The authors refer to "Pink indicates varying degree of LD....". In my version of the manuscript, Figure 1 shows red, white and light blue fields, but no pink fields. Please confirm the accuracy of colors as they show up in the manuscript.

Thank you for pointing this out. The red color has now been clearly stated in the figure caption 1.

Round 2

Reviewer 2 Report

 I prefer do not evaluate this version of the manuscript since I dont agree with the authors answers. 

This manuscript is a resubmission of an earlier submission. The following is a list of the peer review reports and author responses from that submission.

Round 1

Reviewer 1 Report

The study performed association analysis between LEPR gene and reproductive seasonality traits in Rasa Aragonesa sheep. The findings in this study may help us understand the genetic basis of reproductive seasonality traits and provide useful information for maker-assisted selection in chickens. However, the manuscript can be further improved. Generally, the readability should be improved. My suggestions are as follows:

  1. Lines 213-229: It can be tested if there is interaction (epistatic effect) between the two SNPs in MTNR1A and LEPR.
  2. Lines 223-226: What does = stand for in the column of amino acid change in Table 2?
  3. Line 238: Exons were sequenced to detect SNPs. Why SNPs in introns were also found?
  4. Lines 260-262: So, what were the sample sizes for association analyses?
  5. Lines 272-277, 301-309: Were the association analyses conducted twice, using the young and all individuals, respectively?
  6. Lines 315-317: It was said that the SNP in MTNR1A was associated with OCM (lines 315-316), why did it only have a strong influence on TDA trait (line 317)?

Reviewer 2 Report

Comments on the manuscript The LEPR gene is associated with reproductive  seasonality traits in Rasa Aragonesa sheep

The manuscript deals with the association of two genes with seasonality in a sheep breed. In general, the number of individuals analyzed is low, the variations among phenotypes too, and therefore, the results are weak.

Some experimental procedures should be clarified:

One of my questions is regarding the stratification of the population. The authors have corrected the model by the fact that ewes could be related??

Why the authors characterized genes which were already characterized in ewes, but also using only 33 individuals? Any of the SNP discovered was new??

If I understand correctly, 86 ewes showed the same phenotype for P4CM and OCM. Is that correct?? That could explain why only TDA and OCM were the only phenotypes associated to one SNP.

As a rule, I suggest that the authors do not mention any not significant result after Bonferroni correction, since it make the results difficult to follow.

Similar is observed in combined genotype analysis in which a lot of data is provided but most of them is scarcely informative.

In table 4 block 2 for TDA h1 haplotype results are weird. Are the authors sure that 46.3 and 81,3 are in the same statistical group??

In combined analysis, why the authors say that there is only 215 ewes genotyped. The number of null (or positive) genotypes should be stated in each case.

In line 325 the authors stated that ewes showing the C2 genotype decreased the OCM in a 3%. It was also significant??

In table 5, only one combined analysis remains after Bonferroni. Similar results were observed in the rest of the analysis. Since the number of SNP analyzed is low, the results does not seems quite robust. In any case, if the author uses Bonferroni’s (which is correct), they cannot present results with p values after correction of 0.4. This is also noted in the discussion in which the author mentioned the lack if robustness of the analysis and the difficulty to draw valid conclusions. Therefore, I suggest that in case that results are weak, they should be removed from the manuscript.  

Overall, results are not quite well presented. I suggest to the authors a major re-writing focused only on the significant results, thus reducing the amount of information provided in the manuscript.

It is noteworthy that since experimental design and results are not enough solid from my point of view, I did not review introduction nor discussion

Reviewer 3 Report

The authors used polymorphisms in the leptin receptor to investigate a possible association between the leptin receptor gene LEPR and reproductive seasonality in Rasa Aragonesa sheep. In addition, a relationship between the melatonin receptor subtype 1A MTNR1A and LEPR was investigated. Data revealed a link between LEPR and estrous cycle months and concluded that these findings may be used to improve genetic selections for reproductive efficiency in meet production sheep. The manuscript is well written and should be of interest to the reader.

Specific comments:

Line 112, 120 and 124: Please change to “… by Martinez-Royo.”

Lines115-116: Please clarify the two age groups. What does “all:1.9 y” mean. Was there no range in age for this group? Also, the mean +/- SD values should be listed immediately before or after the range and n=155 separately.

Lines 174-176: Please explain why these six SNPs were selected, to clarify for readers without expertise in genomic methods.
